# Candidate Genes for Freezing and Drought Tolerance Selected on the Basis of Proteome Analysis in Doubled Haploid Lines of Barley

**DOI:** 10.3390/ijms21062062

**Published:** 2020-03-17

**Authors:** Magdalena Wójcik-Jagła, Marcin Rapacz, Ewa Dubas, Monika Krzewska, Przemysław Kopeć, Anna Nowicka, Agnieszka Ostrowska, Sabina Malaga, Iwona Żur

**Affiliations:** 1Department of Plant Breeding, Physiology and Seed Science, University of Agriculture in Kraków, Podłużna 3, 30-239 Kraków, Poland; 2The Franciszek Górski Institute of Plant Physiology, Polish Academy of Sciences, Niezapominajek 21, 30-239 Kraków, Poland

**Keywords:** barley, doubled haploid lines, freezing tolerance, drought tolerance, candidate genes, gene expression

## Abstract

Plant tolerance to environmental stress is determined by a very complicated network composed of many intra- and extracellular factors. The aim of this study was to select candidate genes involved in responses to freezing and drought in barley on the basis of previous proteomic studies and to analyze changes in their expression caused by application of both stress factors. Six candidate genes for freezing tolerance (namely the genes encoding elongation factor 1 alpha (*EF1A*), ferredoxin-NADP reductase, a 14-3-3a protein, β-fructofuranosidase, CBF2A and CBF4B) and six for drought tolerance (encoding transketolase, periplasmic serine protease, triosephosphate isomerase, a protein with a co-chaperon region (*GroEs*), pfam14200 and actin) were chosen arbitrarily on the basis of in silico bioinformatic analyses. The expression levels of these genes were measured under control and stress conditions in six DH (doubled haploid) lines with differing freezing and drought tolerance. The results of gene expression analysis confirmed the roles of the candidate genes preselected in this study on the basis of previous proteome analysis in contributing to the differences in freezing and drought tolerance observed in the studied population of DH lines of winter barley.

## 1. Introduction

Plant tolerance to environmental stress is determined by a very complicated network composed of many intra- and extracellular factors. This is why many efforts to improve plant stress tolerance by classical plant breeding methods have achieved only limited success. However, direct and indirect effects of an increasing human population and activity, e.g., climate changes, environmental pollution and decreased biodiversity, have had significant impacts on plant productivity. In the very near future, our understanding of the mechanisms of crop responses to environmental changes will be the most important factor determining food security in many world regions.

Plant responses to various stress factors can be considered on a variety of levels of their organization, beginning with the molecular background, through cells and organs, and ending at the whole plant organism. On the molecular level, plant responses involve a vast number of genes affecting not only plant physiology (such as the regulation of plant hormones, osmotic adjustments and antioxidant defence systems) but also morphology and anatomy (like compositional changes in plasma membranes, leaf rolling and changes in stomatal density and aperture) [1,2,3]. Genomic regulation of plant adaptation to abiotic stress can be examined through genomic and transcriptomic analyses with highly advanced molecular techniques like DNA/RNA-microarrays and next-generation sequencing. However, the huge collections of omics data describing genomic changes induced by particular stress factors need to be properly analyzed and interpreted. Commonly used methods for mining candidate genes are based on: the classic candidate gene approach [4] which matches genes of known sequence and function to QTLs (quantitative trait loci) or MTLs (Mendelian trait loci) [5]; genome-wide scanning, which can use position-dependent, comparative genomics or function-dependent strategies, or a combination of at least two of them; and finally the in silico candidate gene approach, which uses all possible resources obtained from publicly available databases and complex statistics [6]. Candidate gene selection on the basis of proteomic analyses which we present in this study is a rare approach to searching for novel genes possibly involved in complex and quantitative traits, and is the reverse of the methods described above. However, this approach is justified as the distance from a transcript to an active protein is long and many data have demonstrated substantial roles for post-transcriptional and translational processes as well as protein degradation in final protein abundances [7]. 

Six doubled haploid (DH) lines of winter barley examined thoroughly in previous studies with respect to parameters involved in freezing and drought tolerance [8,9,10] were used in the present study. Both of these stress factors impose a water deficit and are considered serious threats to winter barley productivity, especially when they occur at the most critical and sensitive stages, i.e., on young seedlings (freezing) and on plants at the booting stage (drought). The studied DH lines were selected from a population of DHs produced using the anther culture technique from Polish breeding materials and showed increased variation in freezing and drought tolerance levels in comparison with their parental genotypes [8,9,10]. The selected DH lines were examined previously with respect to changes induced by water deficit in the functioning of the photosynthetic apparatus, antioxidative defense and proteome and phytohormone accumulation. 

On the basis of previous proteomic data, several candidate genes possibly involved in plant adaptation to freezing and drought were selected and investigated with respect to transcriptome changes associated with the studied stress factors. These genes encoded transcription factors (CBF4B, CBF2A), and proteins involved in protein synthesis (elongation factor 1 alpha (*EF1a*)), carbohydrate metabolism (β-fructofuranosidase, transketolase, triosephosphate isomerase), the cytoskeleton (actin), redox reactions (erredoxin-NADP reductase), stress responses (20 kDa chaperonin) and cell signaling (14-3-3 protein, pfam14200). The determination of their expression patterns and changes in response to cold and drought treatments in the studied DH lines of barley improves our understanding of their roles in stress adaptation and could provide a basis for more effective classical breeding or engineering strategies leading to increased stress tolerance.

## 2. Results

### 2.1. Candidate Genes Selection

Selection of candidate genes for expression studies was based on the results of previous proteomic studies [8,9]. The proteins potentially involved in the response to each of the tested treatments and differentiating between tolerant and susceptible lines were selected and analyzed in regard to their amino acid sequences. The proteins most likely involved in cold hardening included EPS62279.1 (hypothetical protein M569_12509, partial; cytochrome P450-dependent fatty acid hydroxylase-like), KQJ82088.1 (hypothetical protein BRADI_5g05668, F1 ATP synthase beta subunit-like), EMT12632.1 (ferredoxin-NADP reductase, leaf isozyme), EMT33607.1 (hypothetical protein F775_43926 elongation factor (EF) Tu), KQK13608.1 (hypothetical protein BRADI_1g11290, 14-3-3 protein A like) and XP_013654063.1 (predicted protein—uncharacterized mitochondrial protein AtMg00810-like, retrotransposon like protein) while the proteins potentially involved in drought response included BAJ98295.1 (ferredoxin NADPH cytochrome p450 reductase (CYPOR) leaf isozyme, predicted), BAJ93658.1 (transketolase, predicted), BAK06780.1 (triosephosphate isomerase, predicted), KXG22555.1 (hypothetical protein SORBI_009G237000: trypsin-like serine protease (heat shock, chaperone function, apoptosis)), EMT10427.1 (20 kDa chaperonin, chloroplastic), BAK03652.1 (transport and Golgi organization 2-like protein, predicted), and AAX12161.1 (actin, partial).

Through tblastn analysis of the barley genome, coding sequences with high similarity to these proteins were identified (Table 1). 

Further bioinformatic analysis narrowed down the number of genes for transcriptomic analyses to six related to the response to drought stress and six related to cold hardening. Subsequently, gene-specific primers for RT qPCR were designed:

Cold hardening:1.Elongation factor 1 alpha (EF1A) coding gene (primers designed for the consensus mRNA sequence of: KP293845.1 and KP293846.1);2.Ferredoxin-NADP reductase coding gene (mRNA sequence ID: AK368450.1);3.Gene encoding 14-3-3a protein (mRNA sequence ID: X62388.1);4.Gene encoding β-fructofuranosidase (primers designed on the basis of a consensus mRNA sequence for six splicing variants of MF443751);5.*CBF 4B* (primers designed for the consensus mRNA sequence of: DQ480160.1:7462-8139, DQ445234.1:4551-5228, AY785853.1:86-763 and AY785848.1:84-761);6.*CBF2A* (primers designed for the consensus mRNA sequence of: DQ480160.1:14050-14715, GU461589.1:13-678, GU461588.1:15-680, GU461587.1:21-686 and AY785840.1:13-678).

Response to drought:1.Transketolase gene (mRNA sequence ID: AK362454.1:1-2041);2.Gene encoding periplasmic serine protease (mRNA sequence ID: AK355966.1:49-1332);3.Triosephosphate isomerase gene (mRNA sequence ID: AK375585.1:86-847);4.A protein-coding gene with a co-chaperonin region (*GroEs*) (primers designed for the consensus mRNA sequence of: AK369605.1:156-911 and AK362060.1:215-970);5.Gene encoding pfam14200—ricin-type beta-trefoil lectin domain-like protein (mRNA sequence ID: AK372454.1);6.Actin gene (mRNA sequence ID: AY145451).

### 2.2. Changes in Expression of Selected Genes under Abiotic Stress Conditions

The study revealed different expression levels for the selected genes among barley DH lines, both during hardening to freezing temperatures and in response to drought treatment (Figure 1). However, the observed changes in expression levels were only partly related to stress tolerance.

In the case of hardening, among the studied DH lines, the reactions of freezing-sensitive DH575 and freezing-tolerant (and drought-tolerant) DH602 clearly contrasted with each other (Figure 1). The relative expression levels in these lines were statistically significantly different for all analyzed genes and in the case of three (out of six), *14-3-3a*, *CBF4B* and *ferrodoxin-NADP-reductase*, changes occurred in the opposite direction. With the exception of *EF1*, low-temperature treatment had a strong inhibitory effect on gene expression in DH602. In contrast, four out of six analyzed genes in DH575 were upregulated in comparison with the non-hardened control. The reactions of the other two DH lines (DH158 and DH534) were very similar and intermediate compared with DH602 and DH575 (Figure 1).

The expression of the studied genes was lower in most cases in both non-acclimated and cold-acclimated seedlings of freezing-tolerant plants in comparison with freezing-sensitive ones (Figure 2A). Only in the case of the ferrodoxin-NADP-oxidase and β-fructofuranosidase encoding genes were freezing-tolerant plants characterized by higher expression levels compared with freezing-sensitive lines. However, this was observed only in non-acclimated seedlings.

The relationship between the gene expression patterns of the selected genes and the drought tolerance of the studied DH lines was much more pronounced than that observed for tolerance to cold. The expression of all tested genes significantly decreased in all drought-treated DH lines with greater changes usually occurring (with the exception of *GroEs*) at a higher rate in the drought-tolerant DH lines (Figure 1). The difference in expression levels between tolerant and drought-sensitive lines was visible both in well-watered and drought-treated plants and the expression levels were always higher in drought-tolerant plants (Figure 2B). In five out of six cases the difference in expression level was greater in control plants.

## 3. Discussion

Freezing and drought are major abiotic threats to winter barley productivity, especially when they occur at the most critical and sensitive stages of development—namely, young seedlings and booting plants. Despite many efforts to address these problems, the fact that both freezing and drought tolerance are complex and polygenic traits with strong genotype × environment interactions [11,12] has limited breeding progress. There is still a lack of commercially available varieties that can completely fulfill the requirements of farmers and modern agronomic systems. Fortunately, some biotechnological tools, like DH technology, give the possibility of overcoming this problem. DHs are derived from in vitro cultured haploid cells of male/female gametophytes redirected towards embryogenic development, followed by spontaneous or chemically-induced genome diploidization. The most important advantage of DH technology in comparison with conventional breeding methods is the possibility to achieve complete homozygosity in one generation, allowing a reduction in the time necessary for release and dissemination of new cultivars [13]. Moreover, due to the lack of dominance effects and new pleiotropic or epistatic interactions [14], the whole inherent genetic diversity and potential could be revealed in offspring populations. It is also important that, thanks to this method, plant phenotyping and genotype selection is more accurate and reliable. However, the most efficient plant breeding programs combine DH technology with marker-assisted genomic selection necessary for evaluation and selection of genotypes of interest in each generation. The effectiveness of that combined approach highly depends on the number of available markers [15], which can be increased using various methods, including candidate gene mining.

The six DH lines of winter barley used in this study were selected from DHs produced by androgenesis initiation and they showed increased variation in freezing and drought tolerance in comparison with their parental genotypes [16]. Their thorough examination allowed the identification of several physiological and metabolic parameters associated with stress tolerance acquisition [8,9,10]. In particular, the analysis of proteome profiles provided a basis for the selection of the genes whose expression changes were analyzed in this study [8,9]. Based on the proteome results, we picked six genes potentially involved in the response to cold and six others induced in response to drought.

Interestingly, the selected genes were differentiated only between two of the DH lines identified as freezing-tolerant and freezing-sensitive. The two others, DH158 (freezing-sensitive) and DH534 (freezing-tolerant) did not differ statistically with respect to expression of genes related to cold hardening. Their expression levels were intermediate between those observed for the two other tested lines (which were substantially different). Perhaps DH158 and DH534 display an intermediate reaction in the cold-hardening process between freezing-tolerant and freezing-sensitive lines, and the previously observed differences in their freezing resistance may result from other factors, independent of the changes observed in our experiment. In the case of DH534, which is both freezing- and drought-tolerant, its tolerance could result from its ability to protect tissues against dehydration, which is crucial in both drought and freezing stress, but here was induced only by drought. This means that its protective mechanisms cannot be induced in temperatures above 0 °C, which are used in hardening to freezing but only in response to an actual freezing event. It is also worth mentioning that the expression levels of all the studied genes were lower during cold acclimation in tolerant lines than in susceptible ones, and that the difference in the expression levels between tolerant and susceptible lines decreased during cold acclimation when compared with non-acclimated plants. These results are in line with a recent report for barley in which higher freezing tolerance levels in some accessions were connected with the downregulation of selected genes [17].

Tolerant genotypes react to drought at a higher level of water deficit observed directly inside leaf cells. Because even a slight decrease in hydration can cause a decrease in gene expression levels [18], when hydration was above the reaction threshold a greater decrease in the expression levels of genes not directly related to drought tolerance (e.g., *protease*, which is related to basic metabolism or *triosophosphate isomerase*) or potentially related to drought signal transduction (*pfam14200* coding for lectin) was observed in the tolerant cultivars than in the sensitive cultivars. This was also visible in the comparison of tolerant versus susceptible genotype expression, where the biggest differences were observed in control conditions. After drought treatment, the difference in expression between the tolerant and susceptible lines decreased substantially, probably due to a stronger reaction in tolerant genotypes.

Among the genes identified as possibly associated with the process of cold hardening were genes coding for a factor involved in protein synthesis (EF1A), enzymes that catalyze the electron transfer cascade from photosystem I to NADP+ (ferredoxin-NADP reductase) and the hydrolysis of sugars (β-fructofuranosidase), a regulatory molecule involved in the most important physiological processes (14-3-3a protein) and two transcription factors (CBF2A and CBF4B).

In this group, only CBF genes are known to be involved in plant responses to freezing stress [19,20]. The other genes are not commonly associated with this stress; perhaps the reverse approach to selection of candidate genes could play an important role in revealing less obvious stress response pathways.

For example, *EF1A* is considered a housekeeping gene and is often used as a reference gene in qPCR studies [21,22]. In [21], the EF1A gene showed a low level of expression variation. Similarly, in [20] *EF1* was among the less variable genes. Our results show clearly that expression of *EF1* (both on the transcription and translation levels) is far from stable in response to low-temperature treatment. In fact, the expression level of this gene was lower in tolerant lines both in non-acclimated and acclimated plants.

Lower levels of transcript accumulation in tolerant lines were also observed for the ferredoxin-NADP reductase and β-fructofuranosidase genes. Moreover, the expression levels of these genes decreased substantially after cold acclimation. These results are in accordance with [17] where it was shown and discussed that downregulation of some genes during cold acclimation may contribute to increased freezing tolerance.

Amongst the selected genes related to the drought response, *actin* has been confirmed to play a role in drought responses. According to [23], barley genotypes tolerant to drought are characterized by a lower level of actin gene expression under drought conditions, which was confirmed by the currently obtained results. This lower level of expression may be associated with a higher drought susceptibility threshold.

Among the other genes selected in this study as related to the drought response, *pfam14200* (coding a ricin-type beta-trefoil lectin domain-like protein) may be potentially involved in drought signal transduction. Plant lectins significantly contribute to plant resistance to pathogens by taking part in the perception of environmental signals and their translation into phenotypic response [24].

The direction of changes in transcript accumulation after stress treatment was in most cases not consistent with the direction of changes in the respective protein’s abundance which was previously studied [8,9]. In the present study, accumulation of *14-3-3a* transcripts increased in DH575 and decreased in DH602 after hardening, while the accumulation of the 14-3-3a protein decreased in DH575 and increased in DH602. The direction of change in protein accumulation in DH575 was also different than the change in accumulation of the respective transcripts for EF1 and CBF4B. In DH602, the changes in transcript and protein accumulation after hardening occurred in the opposite direction for all tested genes except *EF1* (Appendix A).

Under drought treatment, the pattern of changes in protein and transcript accumulation was very similar only for the actin and serine protease genes. Accumulation of the co-chaperonin (GroEs) and pfam14200 proteins did not change after drought treatment in any of the tested genotypes, while their transcript accumulation decreased under drought. In drought-treated DH602, the triosephosphate isomerase protein abundance was 12 times higher whereas its transcript accumulation was two times lower. Smaller differences, but in the same direction, were observed in DH435 in the case of the transketolase protein and transcript (Appendix A).

The observed differences in the direction of changes in protein and transcript accumulation after stress treatment may result from various molecular processes. Decreased or unchanged protein abundance combined with increased transcript accumulation in stressed plants may result from translational repression activity of regulatory RNAs [25,26] or protein degradation [27,28]. Conversely, decreased transcript accumulation, when protein abundance either increased or did not change after stress treatment, may be explained by lower stability of transcripts (prone to RNAse degradation). For instance, CBF gene expression levels are highest in the first hours of cold treatment and decrease later on [29], whereas the sampling for gene expression analyses in this study was done when stress symptoms were already visible in the plants. Circadian clock functions have also been described as being affected by mRNA stability [30]. Different patterns of relationships between transcription rate, mRNA level, translation rate and protein level has been reviewed in [31], pointing out that even a slight change in mRNA or protein half-life can alter steady-state levels of mRNA and proteins substantially.

In conclusion, knowledge of changes in the proteome due to drought and cold treatment let us choose candidate genes both with confirmed roles in those stresses and that were not associated with them until now. Gene expression analysis confirmed the roles of the candidate genes preselected in this study on the basis of previous proteome analysis in shaping the differences in freezing and drought tolerance observed in the studied population of winter barley DH-lines. This study also showed that some changes in gene expression can be captured only using comparative proteome-transcriptome analysis. Many of the differences in gene expression observed in this study would be classified as insignificant by transcriptomic-only methods such as RNAseq-based differential expression analysis. In addition to the conclusions drawn directly from this work, the results obtained in this study and the results of previous physiological analyses [8,9] may be summed up in a hypothesis requiring further verification: the suppression of biosynthesis of new proteins, the accumulation of which is related to the level of tolerance, observed both during cold acclimation and drought treatment allows plants to save energy, which can be then used for adaptation to stresses.

## 4. Materials and Methods

### 4.1. Plant Material

The plant material consisted of six DH lines of winter barley produced from breeding materials through the anther culture method based on modified protocols of [32,33] and precisely described by [11]. The selected DH lines were derived from the following parental materials: DH158 (Maybrit ×RAH 983), DH435 (POA 7209/06-3 × RAH 978), DH534 (Traminer × Franziska), DH561 (POA 3574/92/1 × Rosita), DH575 (POA03/260 × Lomerit) and DH602 (cv Souleyka, Saaten Union, Germany). Polish breeding materials were obtained from the DANKO Plant Breeding (Choryń, Poland) and Strzelce Plant Breeding (Strzelce, Poland) companies. The selected DH lines were identified as highly differentiated with respect to their freezing and drought tolerance levels [8,9].

Freezing tolerance tests were conducted in laboratory conditions according to [34] and the ratio of plant survival after freezing treatment (8 h at −12 °C) preceded by 20 days of cold hardening (4/2 °C, day/night; photoperiod 9/15 h; irradiance of 250 μmol m^−2^ s^−1^; SON-T + AGRO) was estimated. Among the tested plant materials, DH534 and DH602 were identified as freezing-tolerant with 80% of plants surviving freezing, whereas DH158 and DH575 were recognized as freezing-susceptible with plant survival not exceeding 40%.

Drought tolerance was estimated according to the procedure of [35] based on the leaf water loss parameter (LWL) according to the following equation:LWL = [(LWC_C_ − LWC_DT_)/LWC_C_] × 100%(1)
where LWC_C_ is the leaf water content of control plants and LWC_DT_ is the leaf water content of drought-treated plants. The LWC parameter was calculated on the basis of leaf fresh mass (LFM) and leaf dry mass (LDM) after 72 h lyophilisation (Freeze Dry System/Freezone 4.5, LABCONCO Kansas City, MO, USA) according to the following equation:LWC = ((LFM − LDM)/LFM) × 100%(2)

In the two drought-tolerant DH lines (DH534, DH561), LWL increased 7.4 and 7.7, respectively, whereas the same parameter in the DH lines recognized as drought-sensitive (DH602, DH435) was almost twice as high (14.7 and 15.2).

### 4.2. Plant Growth and Stress Treatments

Plants were grown in a greenhouse chamber, in plastic pots of 3.7 dm^3^ capacity (six plants per pot) filled with a mixture of soil and sand (1:2, *v*:*v*). Seeds were germinated at a constant temperature of 25 °C for 4 days in the dark. Then, the plants were grown at a temperature of 25/17 °C (day/night), a photoperiod of 12/12 h (day/night) and an irradiance intensity of 400 μmol m^−2^ s^−1^ (HPS lamps, SON-T+ AGRO, Philips, Brussels, Belgium).

At the three–four-leaf stage, the seedlings were subjected to cold hardening/varnalisation (20 days/seven weeks, 4/2 °C, day/night; photoperiod 9/15 h; irradiance of 250 μmol m^−2^ s^−1^; SON-T + AGRO).

After varnalisation the plants were grown in a greenhouse at temperatures of 25–30/18 °C (±2 °C) during the day/night and relative humidity of about 40%.

Soil drought was applied individually for each DH line after full emergence of the flag leaf. The soil water content (SWC) in pots was gradually lowered to 33–35% and was kept at this level for the subsequent two weeks. The SWC of the control pots was maintained at 75–78% by adding appropriate amounts of water every day. The water content in the soil was determined each day between 9:00 and 11:00 a.m. by the gravimetric method [9].

### 4.3. RNA Isolation and Reverse Transcription

The plant material for RNA isolation consisting of 0.03–0.05 g leaf fragments (second leaf in the cold hardening experiment and flag leaf in the drought experiment) was collected under control conditions before hardening, at the stage 13 according to Biologische Bundesantalt, Bundessortenamt and Chemische Industrie (BBCH) scale [36], and under optimal hydration at the beginning of heading (stage 47 according to the BBCH scale), and then after the stress treatment (three weeks of cold hardening/drought). Leaf fragments were collected in three biological repetitions (fragments from three randomly selected plants from a given genotype) under sterile conditions and frozen in liquid nitrogen immediately after collection.

The plant material was homogenized with the addition of carbide balls of 3 mm diameter using a Tissue Lyser (Qiagen, Hilden, Germany) device. The total RNA was isolated with a RNeasy Plant Mini Kit (Qiagen) according to the manufacturer’s instructions. After isolation, the concentration and quality of the RNA obtained for all samples was checked with a UV-Vis Q5009 spectrophotometer (Quawell, San Jose, CA, USA). The reverse transcription reaction was performed with the QuantiTect Reverse Transcription Kit (Qiagen) reagent set according to the manufacturer’s specifications. The obtained cDNA was frozen at −65 °C until it was used as a template in RT-qPCR reactions.

### 4.4. Candidate Genes Selection

Among proteins showing different levels of accumulation in optimal hydration and drought conditions [9] and before and after hardening [8], those with the greatest differences in relation to control conditions were selected. Then, this preselected set was arbitrarily narrowed down to proteins for which a role in the drought response/hardening has been confirmed and proteins that have never been associated with those stresses. The amino acid sequences of the selected proteins were analyzed with the tblastn application [37], searching for nucleotide sequences on the basis of the protein query. The search was narrowed down to sequences from the Hordeum genus. The most similar sequences were selected from the obtained search results. If there were several sequences for the same gene, the consensus sequence was determined using Lalign ([38], to compare two sequences) or Clustal Omega ([39], to compare more than two sequences). Comparisons were made on coding sequences. For each of the two experiments (drought and hardening) six genes were selected for further RT-qPCR studies.

### 4.5. RT-qPCR

Primers for expression experiments were designed using Primer3Plus [40] based on the sequence of a given gene in *Hordeum vulgare* L. or consensus sequences (in the case the sequence of a given gene homologue was not found in barley) (Table 2).

Expression analysis of the selected genes potentially related to drought and cold responses in barley DH lines was performed with quantitative RT-PCR (RT-qPCR) using Real-Time PCR 7500 System (Applied Biosystems, Foster City, CA, USA) in the following steps: UNG activation (50 °C, 120 s) × 1, pre-denaturation (95 °C, 600 s) × 1, denaturation (95 °C, 15 s) and primer linkage and annealing (60 °C, 60 s) × 40, dissociation (95 °C, 15 s followed by 60 °C, 3600 s and 95 °C, 15 s) × 1. The amplification signal was analyzed on the basis of an increase in the fluorescence intensity of SYBRGreen [41].

In the cold hardening experiment, the internal standards were the ADP-ribosylation factor 1-like protein (ADP) and S-adenosylmethionine decarboxylase (sAMD) genes [12], whereas in the drought experiment, the ADP and GADPH genes were used [19]. The reactions were carried out in three biological repetitions for each genotype, each in three instrumental repetitions. Each reaction contained 900 nM of each primer, ca 35 ng of template cDNA and Power SYBR^®^ Green PCR Master Mix (Applied Biosystems).

Relative quantitation of the expression levels of the tested genes was performed on the basis of the modified standard curve method proposed by [42]. The slope of the standard curve, i.e., the dependence of CT values on cDNA concentration in the sample, was used to calculate the amplification efficiency of each gene studied, which was then used to calculate the number of copies of the gene in relation to the reference genes. The number of gene copies in the test samples (after hardening of plants or in drought), normalized in relation to the geometric mean of the number of reference gene copies, was presented as a multiple of the number of copies in the control samples, normalized to the geometric mean of the number of reference gene copies. Geometric means of the number of gene copies were also used for calculation of relative gene expression of tolerant vs. susceptible plants.

### 4.6. Statistical Analysis

Statistical analysis of the results was carried out using the of Statistica 13PL programme (Dell, Round Rock, TX, USA). Standard errors for relative values of normalized gene expression were calculated using all biological and instrumental repetitions. The statistical significance of the differences between lines was determined after logarithmic transformation of the relative value of normalized gene expression for all biological/instrumental repetitions. One-way variance analysis and Tukey’s test for an unequal number of repetitions were used.

## Figures and Tables

**Figure 1 ijms-21-02062-f001:**
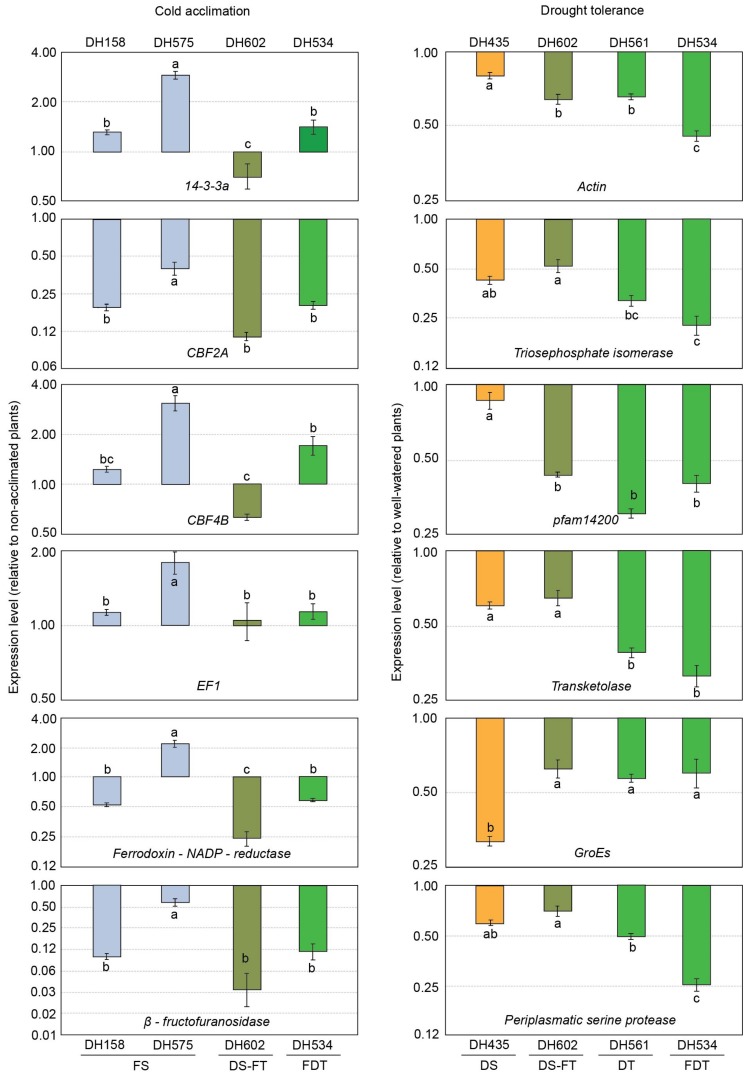
The expression of selected genes during cold acclimation or drought treatment relative to untreated plants in six double haploid lines of winter barley. FS, freezing-susceptible; DS–FT, drought-susceptible–freezing-tolerant; FDT, freezing- and drought-tolerant; DS, drought-susceptible; DT, drought-tolerant. Error bars represent standard error between means of three biological replicates, each with three instrumental repetitions. Analysis of variance was made separately for each gene. Values marked with the same letter do not differ according to Tukey’s test (*p* ≤ 0.05).

**Figure 2 ijms-21-02062-f002:**
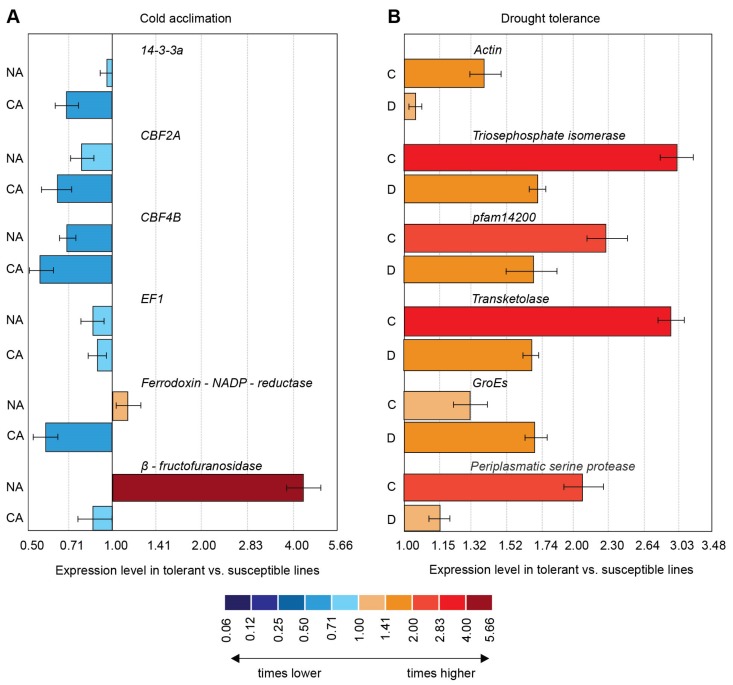
(**A**) Relative expression of genes before (NA) and after cold acclimation (CA) in freezing-tolerant (mean for DH602 and DH534) vs. freezing-susceptible (mean for DH158 and DH575) lines. (**B**) Relative expression of genes in well-watered plants (**C**) and under drought (**D**) in drought-tolerant (mean for DH561 and DH534) vs. susceptible (mean for DH435 and DH602) lines. Error bars represent standard error between the means of two lines (each with three biological replicates and three instrumental repetitions).

**Table 1 ijms-21-02062-t001:** Coding sequences (CDS) obtained through tblastn analysis for selected protein sequences.

Protein ID	Query Cover	E Value	Percent Identity	Coding or Genomic Sequence ID	CDS Annotation
KQJ82088.1	100%	0.0	95.98%	KY636108.1	-
EPS62279.1	54%	1 × 10^−35^	30.96%	JN818424.1	cytochrome P450
EMT12632.1	100%	4 × 10^−180^	94.51%	AK253057.1	-
	100%	4 × 10^−162^	84.71%	AK368450.1	CDS for predicted protein (ferredoxin--NADP(+) reductase)
EMT33607.1	100%	0.0	94.68%	AK250604.1	-
	95%	1 × 10^−33^	29.12%	JN107538.1	*H. brevisubulatum* elongation factor 1 alpha
	95%	7 × 10^−33^	27.60%	KP293846.1	*H. vulgare* eukaryotic elongation factor 1 alpha
KQK13608.1	92%	1 × 10^−173^	100.00%	X62388.1	*H. vulgare* 14-3-3 protein homologue (14-3-3a)
	91%	5 × 10^−152^	86.89%	DQ295786.1	*H. vulgare* subsp. *vulgare* 14-3-3E
	90%	1 × 10^−151^	88.19%	X93170.1	*H. vulgare* Hv14-3-3b
	90%	4 × 10^−151^	87.45%	Y14200.1	*H. vulgare* 14-3-3 protein (Hv1433c)
	91%	1 × 10^−131^	78.75%	DQ295785.1	*H. vulgare* subsp. vulgare 14-3-3D
XP_013654063.1	72%	2 × 10^−34^	53.64%	MF443751.1	putative beta-fructofuranosidase
	73%	2 × 10^−31^	48.65%	AY266442.1	Mla6-2 gene, complete cds
	79%	1 × 10^−21^	41.60%	AF509748.1	*H. vulgare* subsp. *vulgare* Morex barley stem rust resistance protein (Rpg1)
	90%	1 × 10^−37^	48.55%	DQ480160.1	putative glutaredoxin protein, CBF4B and CBF2A
	90%	1 × 10^−37^	48.55%	DQ445234.1	putative glutaredoxin protein, CBF4B and CBF2A
BAJ98295.1	93%	0.0	95.52%	AK368450.1	CDS for predicted protein (ferredoxin−-NADP(+) reductase)
	93%	0.0	95.52%	AK367092.1	CDS for predicted protein (ferredoxin--NADP(+) reductase)
	91%	0.0	77.81%	AK253057.1	-
BAJ93658.1	97%	0.0	100.00%	AK362454.1	partial CDS for predicted protein (transketolase)
BAK06780.1	100%	7 × 10^−159^	89.33%	AK375585.1	-
	100%	3 × 10^−157^	88.93%	U83414.1	*H. vulgare* cytosolic triosephosphate isomerase
KXG22555.1	76%	0.0	93.11%	AK355966.1	periplasmic serine protease, S1-C subfamily
	72%	9 × 10^−85^	50.00%	AK362697.1	periplasmic serine protease, S1-C subfamily
EMT10427.1	100%	7 × 10^−163^	81.13%	AK369605.1	co-chaperonin GroES
	100%	1 × 10^−161^	81.46%	AK362060.1	co-chaperonin GroES
AAX12161.1	100%	5 × 10^−58^	91.75%	AY145451.1	*Hordeum vulgare* actin mRNA, complete cds
BAK03652.1	47%	1 × 10^−74^	68.18%	AK372188.1	region ricin-type beta-trefoil lectin domain-like; pfam14200
	91%	3 × 10^−153^	76.09%	AK372454.1	region ricin-type beta-trefoil lectin domain-like; pfam14200

**Table 2 ijms-21-02062-t002:** Real-time RT-PCR primer sequences.

Gene	Forward/Reverse	Primer Sequence (5′-3′)
*Elongation factor 1 alpha*	forward	TGCCACTTACCCTCCTCTTG
	reverse	TTCTTCTCCACGCCCTTGAT
*Ferredoxin-NADP reductase*	forward	GGCGGGAGAGAAGATGTACA
	reverse	TCAGCCCACACATGTACACA
*14-3-3a protein gene*	forward	TTGGGCTTGCACTCAACTTC
	reverse	GGGAGTCCAGCTCAGCAATA
*β-fructofuranosidase*	forward	CCGACCCTTTGCTCATCAAC
	reverse	GGGTCCCTGAAGTCCTTCTC
*CBF4B*	forward	TTCTCTGGCCTCGCTCTTTC
	reverse	CGCCGCTCTGTTTTACATCT
*CBF2A*	forward	ATGATGCGTGCCTCAACTTC
	reverse	GACGGCGTCCTTGATCTCTT
*Transketolase*	forward	TTGACGAAGGAGGGGAAGAC
	reverse	GGTAGAGCCAGCTTCAATGC
*Periplasmic serine protease*	forward	AAGCGCAAGTTGTCGGATTT
	reverse	CCAGTAGGTCTGCTGACACA
*Triosephosphate isomerase*	forward	AACTCTGAACGCTGGACAGA
	reverse	GCAGTTCTGAGCAGCAACTT
*GroEs*	forward	AGAGGAAACTGCTGGTGGTT
	reverse	CTGCTTCCAGGAGTGATCGA
*Pfam14200*	forward	GTCCCACCCTGTTCTTCTGA
	reverse	CCATGGAGCGCATCAAAGTT
*Actin*	forward	CGACAATGGAACCGGAATG
	reverse	CCCTTGGCGCATCATCTC
*ADP-ribosylation factor 1-like protein*	forward	CGTGACGCTGTGTTGCTTGT
	reverse	CCGCATTCATCGCATTAGG
*S-adenosylmethionine decarboxylase*	forward	TCGGCTACAGCATTGAAGACG
	reverse	CCAAAAACGATATCAGGATGGC
*GADPH*	forward	TTCGGCGAGAAGCCAGTTA
	reverse	CCTCACCCCACGGGATCT

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
