# Peer review of "Candidate Genes for Freezing and Drought Tolerance Selected on the Basis of Proteome Analysis in Doubled Haploid Lines of Barley"

_ijms, 2020, doi:10.3390/ijms21062062_

Round 1
Reviewer 1 Report
The manuscript is further improved after revision.
Author Response
Thank you for your revision. We appreciate your effort very much.
Reviewer 2 Report
The research attempted to utilize prior proteomic analysis of cold and drought response in barley to select candidate genes for studying their expression levels in DH lines of barley, The authors suggest that proteomic analyses may be more representative of stress response than transcriptomic studies because they are further downstream and subject to more interactions. Not sure if this premise is true, however, it is an interesting viewpoint.
The study is well conducted and I only have a few general comments along with some specific editing suggestions.
Most studies have indicated that transcriptomic analyses and corresponding protein levels do not often correspond for multiple reasons, including those mentioned in the introduction, namely, postranscriptional and postranslational modifications, and variations in the half-life of transcripts vs. proteins. Perhaps the authors should say more about this in their discussion.
It was surprising to see that many of the genes were down-regulated in response to cold stress. Was this true in proteomic analysis as well. If so, it would have been instructive to compare the expression of genes corresponding to proteins that either increased or decreased in abundance in the proteome study.
The authors make the point that there approach may help to make progress in breeding varieties that are more stress-tolerant. However, they used DH lines that varied in stress tolerance, so it seems that increased tolerance can be achieved empirically without the level of scrutiny provided in the present study. Also, the correspondence of the selected genes with the stress tolerance of the different DH lines was not perfect, perhaps due to other contributing factors as the authors suggest. Doesn't that argue against the selected approach. Perhaps the authors could add a couple of lines on this subject.
All in all, the study is very interesting and should be of significant interest to breeders and scientists studying stress tolerance in plants. Novel approach and useful information.
Specific edits:
Ln 21 define DH the first time it is used.
Ln 23 change shaping of to contributing to
Ln 31 delete nature
Ln 33 Consider – effects of an increasing human population and activity
Ln 38 Change various levels to a variety of levels
Ln 92 delete determination
Ln 96 change by the to using the
Ln 109 change improves to improved
Ln 122 change whereas to while, change in the drought to in drought
Ln 344 change found to identified (Indicate that it was a blastn analysis of the barley genome.
Ln 356 Consider Subsequently, gene-specific primers
Ln 393 change indicated to revealed
Ln 410 change genes to gene
Ln 552 change on young… to namely, young seedlings and booting plants. Delete, respectively.
Ln 555 change limits to has limited
Ln 556 change agronomy to agronomic
Ln 557 change distinctly to significantly
Ln 558 italicize in vitro
Ln 561 delete whole
Ln 594 change gave to provided, change for selection to for the selection
Ln 850 Change To conclude to In conclusion.
Ln 851 delete arbitrarily (The selection of genes was not arbitrary)
Ln 863 change responses against to adaptation to
Author Response
- Most studies have indicated that transcriptomic analyses and corresponding protein levels do not often correspond for multiple reasons, including those mentioned in the introduction, namely, postranscriptional and postranslational modifications, and variations in the half-life of transcripts vs. proteins. Perhaps the authors should say more about this in their discussion.The paragraph concerning differences between transcript and protein accumulation was broadened, according to the Reviewer’s suggestion.
- it would have been instructive to compare the expression of genes corresponding to proteins that either increased or decreased in abundance in the proteome study.A supplementary Table was added to the paper. It contains a comparison of changes in transcripts and proteins accumulation after cold acclimation and drought treatment and is referred to in the discussion section of the revised manuscript.
- The authors make the point that there approach may help to make progress in breeding varieties that are more stress-tolerant. However, they used DH lines that varied in stress tolerance, so it seems that increased tolerance can be achieved empirically without the level of scrutiny provided in the present study. Also, the correspondence of the selected genes with the stress tolerance of the different DH lines was not perfect, perhaps due to other contributing factors as the authors suggest. Doesn't that argue against the selected approach. Perhaps the authors could add a couple of lines on this subject.We broadened the fragment about importance of DH technology in breeding, and the effectiveness of combining this technology with marker-assisted breeding.
- Specific edits
All of the edits pointed out by the Reviewer were corrected according to the suggestions.
This manuscript is a resubmission of an earlier submission. The following is a list of the peer review reports and author responses from that submission.
Round 1
Reviewer 1 Report
The manuscript can add to the knowledge in this area, as authors reported candidate genes related to cold and drought stress previously not reported. The text is concise and understandable written. I recommend a few minor improvements:
Introduction:
morpho-physiological basis of cold and drought tolerance should be mentioned in introduction
Plant Material:
Although DH lines are described for their cold and drought tolerance elswhere, here should be mentioned what criteria was used to classified DH lines as drought or cold stress tolerant/sensitive
RNA isolation and reverse transcription:
Define at what stage of plant growth (using Zadoks or other appropriate scales of plant growth stages) leaf sampling for drought and cold treatment was applied
Discussion:
It would be interesting to relate this findings with agronomic features of selected DH lines if available (are they just stress sensitive/tolerant or they have some agronomical value/high yield-how gene expression is related to agronomic value of the genotype).
Reviewer 2 Report
The manuscript deals with the identification of candidate genes related to freezing and drought tolerance using insilico approaches with gene expression analysis. However, the present work has to be improved with additional experiments and should be validated in more number of lines before consideration for publication.
The six genes selected from the previously published proteome results is fine but it can not be concluded with the mere gene expression analysis in very few DH lines. Also most of the proteins selected are not directly related to stress response and among them only few have displayed the difference in the gene expression in the DH lines studied. In addition, the outcomes of the present work can be of interesting only if the genes are validated in more number of lines with contrasting tolerant traits.
Discussion Section needs to be improved (for example:179-180).